# Investigation of Body Composition and Cardiac Sports Adaptation in Elite Water Polo Players

**DOI:** 10.3390/sports13060180

**Published:** 2025-06-09

**Authors:** Mark Zamodics, Mate Babity, Gusztav Schay, Agnes Bucsko-Varga, Eva Kovacs, Marton Horvath, Kinga Grebur, Marcell Janos Laszlo, Alexandra Fabian, Balint Karoly Lakatos, Szilvia Herczeg, Hajnalka Vago, Attila Kovacs, Bela Merkely, Orsolya Kiss

**Affiliations:** 1Heart and Vascular Center, Semmelweis University, 68 Varosmajor Street, 1122 Budapest, Hungary; babity.mate@semmelweis.hu (M.B.);; 2Department of Sports Medicine, Semmelweis University, 68 Varosmajor Street, 1122 Budapest, Hungary; 3Department of Biophysics and Radiation Biology, Semmelweis University, 34-37 Tuzolto Str., 1094 Budapest, Hungary

**Keywords:** water polo, body composition, mix sport, athlete’s heart, sports medicine

## Abstract

The effects of physical activity on skeletal muscle mass and cardiac function are well-documented, but there is limited information on the relationship between the two. Furthermore, differentiating between the ‘athlete’s heart’ and pathological cardiac conditions often presents challenges. We aimed to analyze resting echocardiographic parameters in elite water polo athletes, considering sex, anthropometrics, and body composition. We examined 161 youth and adult athletes (age: 19.7 ± 5.6 years, male: 50.9%). Data analysis was performed with R software (version 4.2), using multivariate linear regression models. Confounders besides the main predictor were sex, age, and height. Male players had higher weight (87.55 ± 12.83 vs. 69.77 ± 9.8 kg), height (188.59 ± 6.82 vs. 173.47 ± 6.76 cm), skeletal muscle mass (SMM, 43.87 ± 5.50 vs. 30.38 ± 3.95 kg), and fat-free mass (FFM, 76.60 ± 9.23 vs. 54.52 ± 6.68 kg) and lower percentage of body fat values (12.14 ± 4.00.vs 21.51 ± 4.76%) compared to the female players. Youth players had lower height (178.51 ± 9.53 vs. 186.74 ± 9.27 kg) and weight (74.34 ± 12.12. vs. 88.23 ± 14.66 kg) compared to adults. Left ventricular end-diastolic and end-systolic diameters correlated positively with SMM (Est: 0.38, StE: 0.08, *p* < 0.001 and Est: 0.42, StE: 0.11, *p* < 0.001) and FFM (Est: 0.25, StE: 0.05, *p* < 0.001 and Est: 0.25, StE: 0.06, *p* < 0.001). Right ventricular end-diastolic diameter correlated positively with SMM (Est: 0.18, StE: 0.08, *p* < 0.05) and FFM (Est: 0.12, StE: 0.05, *p* < 0.05). Interventricular septal wall thickness showed positive correlation with SMM (Est: 0.16, StE: 0.04, *p* < 0.001) and FFM (Est: 0.10, StE: 0.02, *p* < 0.001). Left ventricular posterior wall thickness correlated with SMM, with a stronger correlation in females (Est: 0.17, StE: 0.05, *p* < 0.001) than in males (Est:0.7, StE: 0.04, *p* < 0.05). The close relationship between body composition and cardiac dimensions provides an opportunity for professionals to distinguish between athlete’s heart and pathological conditions.

## 1. Introduction

The medical and scientific support of elite athletes has become essential over the past decades. Regular exercise leads to the expansion of heart chamber dimensions, thickening of the myocardium, and specific functional adaptations, collectively resulting in the development of the so-called “athlete’s heart” [1]. Despite intensive scientific research, however, the border between “athlete’s heart” and specific pathological disorders like hypertrophic cardiomyopathy, arrhythmogenic cardiomyopathy, or dilated cardiomyopathy still remains unclear in certain cases [2]. These dilemmas are strongly connected with the fact that structural and functional alterations highly depend on the type of sport, training intensity, sex, age, and anthropometric parameters [2,3].

Previous research has already dealt with the problem of differential diagnosis between “athlete’s heart”, hypertrophic, and arrhythmogenic cardiomyopathy [4,5]. Research highlights both the importance and the challenges of differential diagnosis, while emphasizing the significance of training intensity, type of sport, sex, age, and anthropometric parameters. Studies consistently agree on the need for sport-specific, individualized assessment.

Combining these basic parameters of an athlete with some widely available resting measurements of physical fitness like body composition analysis data could help specialists differentiate between sport adaptation changes and pathologies [6]. Bioimpedance body composition analysis is a technique often applied in sports facilities and medical centers as well; its application is easy, reliable, and reproducible. The method operates through the determination of impedance between the limbs, wherein this value initially aids in determining total body water. Through further calculations, the instrument estimates the quantity of muscle and fat mass, broken down into segments for the limbs [7]. Different sports activities cause distinct changes in body composition, and subtle changes caused by altered training intensity, diet, etc., can also be monitored regularly by body composition analysis [8]. As the type of sport and training intensity influence body composition, changes in body composition—such as the increase in muscle mass—are important features and determinants of maximal performance [9,10]. Moreover, the changes in body composition in youth athletes are part of normal physical development [11]. As a youth athlete’s body is under extreme physical load during regular high-intensity trainings, optimal nutrition is necessary, and body composition measurements help to follow up physiological development [8].

Water polo is a mixed sport, which requires adequate endurance, good dynamic and static performance, as well as specific skills. It is one of the most successful and popular sports in Hungary, and our national youth and adult teams regularly end up among the top performers in international competitions. Although we already have some data about the cardiac sport adaptation of water polo players, to best of our knowledge, we do not have any information about correlations between basic resting parameters and “athlete’s heart” adaptation in these athletes [12,13]. Recently, a cooperation has been established between the Heart and Vascular Center of Semmelweis University, Budapest, Hungary, and the Hungarian Water Polo Federation to ensure optimal cardiology and physical fitness follow-up of elite water polo athletes, members of the national teams from youth players to the adults. Within the confines of this cooperation, we had a unique opportunity to carry out comprehensive resting and exercise examinations of elite water polo athletes aged 13 and above (including adults).

As the literature lacks sport-specific, sex- and age-adjusted echocardiographic parameters, the aim of our research was to compare the results of body composition and echocardiographic assessments conducted on elite youth and adult water polo players. Furthermore, based on these findings, we intend to develop a free, online calculator that provides reference ranges adjusted for sex, age, and body composition, thereby supporting personalized diagnostics and clinical decision-making.

## 2. Materials and Methods

### 2.1. Study Design and Experimental Protocol

This study was designed as a single-center, cross-sectional observational study, conducted at the Heart and Vascular Center of Semmelweis University between 1 September 2021, and 31 August 2022. The primary aim was to perform a detailed cardiological and physiological assessment of elite male water polo players and to analyze the relationship between body composition and echocardiographic parameters. As a secondary objective, we developed a web-based calculator to provide a rapid, user-friendly tool for estimating echocardiographic parameters based on body composition data.

Using purposive sampling, we performed an extended cardiological screening of Caucasian elite water polo players. As part of the screening process, participants underwent medical history assessment, blood testing, blood pressure measurement, body composition analysis, resting electrocardiogram (ECG), and cardiopulmonary exercise testing (CPET).

The inclusion criteria included a training volume exceeding 10 h per week, current or former membership of a national team, absence of symptoms, no known cardiovascular disease, and engagement in regular high intensity training during the past month.

Exclusion criteria were the presence of symptoms at the time of examination, newly detected ECG abnormalities, or cardiac ultrasound findings deviating from international normative standards without having undergone detailed diagnostic follow-up.

It is important to highlight that athletes with known ECG or echocardiographic changes were only allowed to participate in the study if previous investigations—such as cardiac MRI, cardiac CT, or genetic testing—had ruled out any underlying pathology. Elite athletes have been regularly examined at the Heart and Vascular Center since 2008, and the majority of the water polo players included in this study had undergone multiple previous evaluations at our clinic. A significant deviation from their earlier results was also considered an exclusion criterion. A total of 168 individuals participated in our study, of whom 7 were excluded (1 due to the presence of symptoms, 3 due to ECG abnormalities, 1 for not training regularly in the recent period, and 2 due to echocardiographic abnormalities).

In total, 161 elite Caucasian youth and adult national team member water polo players completed the study (age—19.7 ± 5.6 years; age range—13.8–37.0 years; youth player—67.7%; male—50.9%; training—18.4 ± 6.1 h/week).

Based on their age, the players were assigned to youth (≤20 years) or adult (>20 years) female and male groups according to international World Aquatics (previously FINA) regulations [14]. Prior to the study, for all athletes under 18-years-old, a parent and/or legal guardian gave written informed consent to the examinations. Scientific and Research Committee of the Medical Research Council approved the study (No.: IV/10282-1/2020/EKU) according to the Ethical Guidelines of the Helsinki Declaration and to Good Clinical Practice. All measurements were performed at least 18 h after the last training session.

### 2.2. Cardiology Screening

All athletes underwent a thorough cardiology screening, including a detailed questionnaire, physical examination, body composition analysis, resting 12-lead ECG, blood pressure measurement, laboratory examinations, echocardiography, and cardiopulmonary exercise testing. Athletes with pathological findings that could affect the results were excluded from the study. All examinations were supervised by a cardiology and sports medicine specialist. The examinations were conducted on weekdays between 8:00 AM and 12:00 PM.

### 2.3. Body Composition Analysis

Body composition analyses were carried out via bioelectrical impedance measurements (InBody 770, InBody Co. Ltd., Seoul, South Korea) [15]. Due to the recommendations, body composition measurements were performed after a minimum of 3 h of fasting. Otherwise, athletes were instructed to follow their regular diet before the examinations. All jewelry, watches, and clothing were removed, except for underwear and bras. Participants were required to urinate at least 30 min prior to the measurement and were instructed to avoid showering or using a sauna within 3 h before the assessment [16].

Detailed analyses of skeletal muscle mass (SMM), percent body fat (PBF), fat-free mass (FFM), and body fat mass (BFM) were carried out.

### 2.4. Echocardiographic Measurements

Routine resting echocardiography was performed, following the current guidelines [17]. Standard two-dimensional long and short axis, as well as M-mode images were acquired using a 4Vc- Probe (GE Healthcare, Oslo, Norway) connected to a commercially available system (Vivid E95, GE Healthcare, Norway). A one-lead ECG was continuously displayed during the examination. Basic resting echocardiographic parameters: left ventricular end-systolic and end-diastolic diameters (LVESD and LVEDD, parasternal long axis view), right ventricular end diastolic diameters (RVEDD), left and right atrial (LA and RA) length and width (apical 4 chamber view), as well as interventricular septal wall thickness (ISWT) and left ventricular posterior wall thickness (LPWT) in diastole (parasternal long axis view) were measured and averaged over three consecutive cycles. Echocardiographic parameters were correlated with anthropometric data (age, sex, height), training hours, and the body composition measurements detailed above.

### 2.5. Statistical Analysis

Data analysis was performed with the R statistics software (Version 4.2., R Core Team (2017). R: A language and environment for statistical computing. R Foundation for Statistical Computing, Vienna, Austria. https://www.R-project.org/, accessed on 1 September 2021) using multivariate linear regression models.

Besides the main predictor variable, the models included three possible confounders in every analysis case: the sex, age, and height of the patients. The overall correlation coefficient was adjusted to account for the number of ordinal and numeric variables in the model.

Descriptive data are represented as mean ± SD. For variables where the sex effect was significant, the average values are represented as mean ± SD of the effect on the specific parameter. Correlations are characterized as Estimate (Est): the estimated change in Y for a one-unit change in X and with standard error (StE). Statistical significance was determined as *p* < 0.05.

## 3. Results

### 3.1. Study Population and Basic Parameters

Basic anthropometric and body composition parameters are listed in Table 1.

Regarding the anthropometric parameters, the adult male players had higher weight, height, SMM, and FFM and lower PBF values compared to the adult female players. No significant difference was found in BFM between the two adult groups. Youth players had lower height and weight in both sexes compared to adults. Adult male athletes had higher SMM, FFM, BFM, and PBF values compared to youth males, while adult female athletes, similarly to males, had higher SMM and FFM values than youth female players. However, no significant difference was found between the fat parameters (PBF, BFM) of youth and adult female players. Comparing the youth groups, youth male players had higher height, weight, SMM, and FFM and lower BFM and PBF than youth female players. However, a mean age difference could also play a role in these results (Table 1).

Resting basic echocardiographic parameters are shown in Table 2.

The adult male players had larger ventricular (LVEDD, RVEDD) and atrial cavity dimensions (RA and LA width and length) and had greater ISWT than adult female players. No differences in LPWT were found between the adult groups. Adult male players had larger ventricular (LVEDD, RVEDD) and atrial cavity dimensions (RA and LA width and length) and larger myocardial thickness (ISWT, LPWT) than youth male players, but this was not the case for LVESD. As for female athletes, adult players had larger LA width, ISWT, and LPWT than youth female athletes. However, no significant difference was found in ventricular dimensions (LVEDD, LVESD, and RVEDD) and other atrial cavity dimensions (RA length and width and LA lengths) between adult and youth female athletes. Youth male players had larger ventricular cavity sizes (LVEDD, LVESD, and RVEDD) and greater myocardial thickness (ISWT and LPWT) than youth female players. Although youth male players had a larger RA width, there were no other differences in the atrial diameters between the two youth groups.

### 3.2. Correlations Between Basic Parameters and Body Composition

Skeletal muscle mass correlated with height (Estimate (Est, the estimated change in Y for a one-unit change in X): 0.42, standard error (StE): 0.04, *p* < 0.001) and age (Est: 0.22, StE: 0.05, *p* < 0.001). Male players with the same height and age had higher SMM (by an average of 6.76 ± 0.77 kg) compared to female players. Fat-free mass correlated with height (Est: 0.69, StE: 0.07, *p* < 0.001) and age (Est: 0.44, StE: 0.08, *p* < 0.001) and male players of the same height and age had higher FFM (by an average 10.86 ± 1.26 kg) compared to females. Body fat mass increased with age (Est: 0.21, StE: 0.08, *p* < 0.01) and had a positive tendency with height (Est: 0.12, StE: 0.06, *p* = 0.06). In the same age and height groups, females had higher BFM (by an average 6.48 ± 1.16 kg). In total, height and age had no significant effect on PBF, while in terms of sex, males had less body fat percentage (by an average of 9.29 ± 1.09%). However, we found a positive correlation between PBF and age in the male population (Est: 0.32, StE: 0.09, *p* < 0.001), while in females, age had no obvious influence on PBF. We found no correlation between weekly training hours and body composition parameters.

### 3.3. Correlations Between Echocardiographic Parameters and Body Composition

In the following paragraphs, the correlations between body composition and echocardiographic parameters are detailed. The SMM and echocardiographic correlations are illustrated in Figure 1, and in the case of FFM, in Figure 2

### 3.4. Left Ventricular End-Diastolic Diameter

Left ventricular end-diastolic diameter correlated with SMM (Est: 0.38, StE: 0.08, *p* < 0.001) and FFM (Est: 0.25, StE: 0.05, *p* < 0.001), while additional confounders (height, age, sex) had no significant effect on it. Moreover, LVEDD also correlated with BFM (Est: 0.14, StE: 0.06, *p* < 0.02), but additional confounders like height (Est: 0.12, StE: 0.05, *p* < 0.05) and sex (Est: 3.97, StE: 0.93, *p* < 0.001) also affected it. Male players of the same BFM, age, and height had higher LVEDD (by 3.97 ± 0.93 mm on average) than female players. Age, as an additional confounder, did not affect LVEDD. No correlation was found between LVEDD and PBF.

### 3.5. Left Ventricular End-Systolic Diameter

Significant correlation was found between LVESD and SMM (Est: 0.42, StE: 0.11, *p* < 0.001), and also between LVESD and FFM (Est: 0.25, StE: 0.06, *p* < 0.001), while additional confounders (height, age, or sex) had no significant effect on it. In our study, BFM and PBF had no influence on LVESD.

### 3.6. Right Ventricular End-Diastolic Diameter

When correlating RVEDD, body composition, age, sex, and height, we found a positive correlation between RVEDD and SMM (Est: 0.18, StE: 0.08, *p* < 0.05), as well as between RVEDD and FFM (Est: 0.12, StE: 0.05, *p* < 0.05), while additional confounders (age, height, sex) did not influence RVEDD. In male players, a correlation was found between BFM and RVEDD (Est: 0.23, StE: 0.1, *p* < 0.05), and height was also an influential additional confounder (Est: 0.13, StE: 0.04, *p* < 0.01).

### 3.7. Right Atrial Width and Length

Regarding the atrial dimensions, RA width and RA length correlated with SMM (respectively, Est: 0.42, StE: 0.14, *p* < 0.01 and Est: 0.45, StE: 0.15, *p* < 0.01) and FFM (respectively, Est: 0.26, StE: 0.08, *p* < 0.05 and Est: 0.29 StE: 0.09 *p* < 0.05), while the additional confounders had no influence.

### 3.8. Left Atrial Length and Width

Moreover, LA lengths correlated with SMM (Est: 0.44 StE: 0,14 *p* < 0.01), and FFM (Est: 0.28, StE: 0.09, *p* < 0.01), while sex was an influential additional confounder (transition to male, SMM: Est: −3.93, StE: 1.17, *p* < 0.05, FFM: Est: −3.94, StE: 1.63, *p* < 0.05). Based on these findings, women exhibit, on average, 3.93 mm larger LA lengths compared to men of the same age, height, and skeletal muscle mass. A significant correlation was found between LA length and BFM in male players, while height was a strong influential confounder (respectively, Est: 0.40, StE: 0.18, *p* < 0.05 and Est: 0.26, StE: 0.08, *p* < 0.001). Left atrial width showed no correlation with body composition parameters.

### 3.9. Interventricular Septal Wall Thickness

In the examined group, ISWT showed positive correlation with SMM (Est: 0.16, StE: 0.04, *p* < 0.001) and with FFM (Est: 0.10, StE: 0.02, *p* < 0.001), while additional confounders had no significant effect. Although BFM also positively correlated with ISWT (Est: 0.08, StE: 0.03, *p* < 0.01), sex was a manipulator additional confounder (transition to male, Est: 1.14, StE: 0.41, *p* < 0.01). Male players with the same BFM, age, and height had greater ISWT (by 1.14 ± 0.41 mm on average).

### 3.10. Left Ventricular Posterior Wall Thickness

Higher SMM and FFM values were associated with higher LPWT. These correlations were stronger in female players (SMM: Est: 0.17, StE: 0.05, *p* < 0.001, FFM: Est: 0.01, StE: 0.03, *p* < 0.001) than in male players (SMM: Est: 0.01, StE: 0.04, *p* < 0.05, FFM: Est: 0.04, StE: 0.03, *p* < 0.05) and sex was a manipulator additional confounder in both SMM and FMM cases (transition to male, Est: 3.63, StE: 1.55, *p* < 0.05, Est: 3.98, StE: 1.63, *p* < 0.05). Age and height did not influence the above correlations.

### 3.11. Online Calculator for Basic “Athlete’s Heart” Parameters

Synthetizing our resting body composition analysis parameters, echocardiographic measurements, and complex statistical analysis data, we developed an easy-to-use online calculator (SEMMELWEIS ATHLETE’S HEART CALCULATOR), which is freely accessible to every sports professional on the website of Semmelweis University Heart and Vascular Center [18]. The calculator uses the obtained research results based on multivariate linear regression. By entering basic parameters of an athlete (age, sex, height, weight), and, optionally, body composition analysis data (either SMM or FFM), the program calculates percentiles of basic resting echocardiographic parameters (LVESD, LVEDD, RVEDD, RA length, RA width, LA length, LA width, ISWT, and LPWT) one by one, and also provides percentile graphs including the individual athlete with signs on them. While significant associations were characteristically found with body composition only, confounders such as age, height, and sex did not prove statistically significant, as detailed in the results. However, the calculator also takes into account these non-significant associations, as in the multilinear regression model, they contribute to the refinement of the estimation. Currently, the details of the computation processes behind the Semmelweis Athlete’s Heart Calculator site are under patent process in Hungary.

https://semmelweis.hu/varosmajor/en/reasearch/sportcardiology/heart-calculator/, accessed on 1 September 2024.

The results help to action decision-making using the percentile numbers if the measured parameter falls into the normal range, or potentially pathological range (over 95th percentile or under 5th percentile). In the case of potential pathological parameters, the program recommends a further cardiac evaluation of the athlete. In the future, the database which the calculations are based on can be easily updated with our new measurement data of more water polo athletes as well as with new data of practitioners of other sports.

## 4. Discussion

Our basic examinations revealed a significant difference in terms of anthropometric and echocardiographic parameters between male and female athletes and also between young and adult athletes. Adult male players had larger ventricular and atrial cavity dimensions and had greater ISWT than adult female players. Moreover, for almost all examined basic echocardiographic parameters (except LVESD), adult male players had higher values than youth male players in our results. In case of female athletes, less significant differences in echocardiographic parameters were found between the age groups, although the adult female players had greater ISWT, LPWT, and LA values than youth female players. These results confirm the fact that we cannot use the same echocardiographic parameters for male and female athletes, and neither can we use echocardiographic parameters defined for normal “athlete’s heart” in adults in case of youth athletes. Up to now, however, normal parameters for different sexes and for youth athletes have unfortunately been missing from the literature.

Personalized echocardiography, relative to body dimensions, is already used in pediatric cardiology practice [19]. Lopez et al. created a Z score calculator, which determines Z scores by body surface area (BSA), while age, sex, race, and ethnicity are a non-determining part of the model [20]. The program is widely used in pediatrics, but it is less useable for athletes, as the BSA is less accurate than the FFM [21] and, due to exercise-induced structural changes in the heart, youth athletes easily fall outside the normal range [22].

The shortcomings of the clinical evaluation of our athletes are thought-provoking, as it is well known that (especially genetic) cardiac muscle diseases first induce structural cardiac changes in these ages. Therefore, we need to do our best to find a solution for optimal cardiology screening and to establish normal values for these young athletes.

By our results, SMM and FFM correlated with age, sex, and height. Given that the studied population included many young, developing athletes who regularly gain muscle mass due to both hormonal changes and continuous training, our results are not surprising and are well in line with earlier reports [23,24]. While previous research established the relationship between the amount of time spent on training weekly and body composition, we could not demonstrate this connection [25]. This is likely due to different training intensities and due to the mixed nature of water polo; that is, highly different amounts of dynamic and static exercise at different sports clubs where members of the national teams train during the year.

In general, fat parameters showed less correlation with the anthropometric data in our elite water polo athlete population. BFM increased with age and showed a nearly significant correlation with height. The increase in BFM with age and the increase in PBF with age in male athletes can also be explained by the studied population; due to growth and neuroendocrine changes in youth athletes, fat mass increases in parallel with muscle mass.

The detected higher SMM and FFM and lower PBF and BFM in male players as compared to female athletes are also due to hormonal and genetic differences between the sexes [26].

Our results refer to a strong correlation between LVEDD, LVESD, ISWT, and LPWT and SMM and FFM. As training volume (the quantity, quality, and years spent in regular training) increases, skeletal muscle mass and other components of fat-free mass (such as bone mass) increase. Expectedly, cardiovascular adaptation should follow the above skeletal changes to provide optimal oxygen and metabolic supply for the increased demand. In the case of mixed sports like water polo, the increase in left ventricular chamber size and the increase in septal and posterior wall thickness (as a sign of cardiac muscle mass increase) are well known signs of athletic cardiac adaptation [27]. In the case of LPWT, sex was an influential additional confounder. Female athletes had greater LPWT changes for the same body composition alteration. According to previous research, male and female exercise-induced cardiac remodeling are considered similar, although female athletes show quantitatively less physiological transformation than their male counterparts [28,29]. Based on our study, although the absolute values of sport adaptation are more expressed in males, some signs of remodeling (like LPWT) show greater changes to SMM and FFM increase in female athletes compared to males. In an animal model of sport adaptation, Olah et al. also found a more pronounced relative cardiac sport adaptation in female animals compared to males [30].

The strong involvement of the right ventricle in cardiac sport adaptation has also been revealed recently [31,32]. Therefore, the correlation that was found in our study between right ventricular end-diastolic diameter, SMM, and FFM is not surprising either. Moreover, the remarkable increase in atrial diameters is also a common indicator of “athlete’s heart” [33,34]. In our study, as part of cardiovascular sport adaptation, right atrial length and width and left atrial with also correlated with SMM and FFM.

In general, similarly as in the context of anthropometric parameters, fat parameters showed less correlation with the echocardiographic data. Regarding BFM and the ventricles, we found that BFM was only associated with LVEDD, but in this case, height and sex were also influential confounders. PBF did not show correlation with the chamber sizes. Similar results were obtained for atria; only one correlating parameter was found (RA length and BFM). These results can be explained by the fact that body fat content depends on various other factors besides sport activity; although regular training decrease BFM and PBF, other conditions like the diet or genetic factors also affect these parameters to a high degree. Moreover, being practitioners of a mixed water sport, elite water polo athletes usually do not have a remarkably low body fat mass and percentage.

Previously, Pressler et al. researched the association of body composition and left ventricular dimensions in elite athletes [21]. Athletes over the age of 18 who performed dynamic exercise took part in their research and they used the skinfold measurement technique for body composition measurement. In their case, FMM and left ventricular dimensions were closely related, which is in accordance with our research. In this study, however, the correlation between body surface area and left ventricular dimensions was considerably weaker.

Furthermore, Mascherini and colleagues examined the relationship between the left ventricle and FFM in young athletes [35]. Using the skinfold measurement technique for body composition assessment, they found a close correlation between FFM and left ventricular dimensions. Although the research used the less precise skinfold measurement technique and did not precisely determine the sport and weekly sports intensity, the results are consistent with previous findings. Furthermore, the research has shortcomings as it only conducted the study in one heart chamber (the left ventricle), while sports adaptation affects all chambers of the heart.

Also, in good agreement with our research, D’Ascenzi and colleagues have recently confirmed that changes in left ventricular mass calculated by echocardiographic measurements are in close association with changes in dual-energy X-ray absorptiometry measurements of FFM in elite athletes, suggesting that “athlete’s heart” develops parallel with the increase in the metabolically active muscle tissue induced by training [36]. However, this research involved only a low number of adult male soccer athletes.

Water polo is a mixed sport, and previous research has also confirmed that it involves both aerobic and anaerobic exertion [37,38]. In our study, the results of cardiac ultrasound and body composition measurements separately showed observable changes caused by aerobic and anaerobic training, and the strong correlations found in the comparison of the results are largely due to this. Only the highest-level (member of the national team, top division) water polo players participated in our study, where, due to the high training load, both cardiac and skeletal muscle adaptations are more pronounced, contributing significantly to our findings. Genetics undoubtedly plays an important role in determining who becomes an elite water polo player, but the associations observed in our study are likely attributable to the high training volume. Further research is needed to better understand the role of genetics in this context.

Although the three studies mentioned above differ in their details and design, they had similar results to our research [21,35,36]. Therefore, four independent studies using three different methods confirmed that body composition, especially FFM, is closely related to heart echocardiographic parameters.

Previous research has described a positive correlation of body surface area with cardiac cavity size. Our research has examined these correlations further and found good correlations between body composition parameters and echocardiographic data in elite water polo players [39]. Correlations between body composition and cardiac parameters could be detected in both the right and left sides of the heart, both in terms of cavity sizes and wall thickness values. Based on our results, echocardiographic parameters of sport adaptation could be predicted better by also considering the results of body composition analysis. Applying our results, we developed the Semmelweis Athlete’s Heart calculator, utilizing the findings of the research to provide sport-specific normal echocardiographic values also taking into account sex, age, height, and body composition parameters. This tool facilitates the differentiation between pathologies and signs of sport adaptation, particularly for facilities with limited imaging modalities and less financial resources. While cardiac MR stands as a gold standard method for confirming structural heart diseases, it may be less accessible or unavailable in economically challenged countries [40,41]. Therefore, additional diagnostic options become particularly crucial in these cases. It is essential to emphasize that the online calculator is not designed to replace cardiac MRI but serves as an additional diagnostic aid, potentially reducing the need for unnecessary examinations and the costs of evaluation.

## 5. Conclusions

The aim of our study was to find simple, sensitive, and easily available resting parameters of sport adaptation to help differentiate between “athlete’s heart” and cardiac diseases in elite athletes. We examined commonly used bioelectrical impedance body composition parameters as sensitive markers of sport adaptation changes in the whole body, as well as basic anthropometric data, and correlated them with basic and widely available echocardiographic parameters throughout the examination of the Hungarian youth and adult national water polo teams.

Our results show that the correlations between FFM and SMM and echocardiographic sport adaptation parameters seem to be much stronger than the correlation between height and echocardiographic measures. Previous research has described that widely used indexation for body surface derived from height seem to have less clinical value than the above-mentioned body composition parameters in case of cardiac examinations of elite athletes [21,42].

In the light of our results, we can interpret basic resting echocardiographic data in a personalized way with the help of body composition parameters in elite athletes. Based on our results, we have created a freely accessible online calculator, which helps the differentiation process between normal “athlete’s heart” and cardiac pathologies by giving percentile values for resting echocardiographic parameters calculated for sex, age, height, and body composition [18]. Applying the results of our study, we may get closer to appropriately interpreting these parameters in a more patient-centralized manner, which helps in diagnostic and therapeutic decision-making.

Our findings contribute to a more holistic approach for physicians and trainers, supporting a broader understanding of the athlete’s heart. Furthermore, the use of the Semmelweis Athlete Heart Calculator enables the development of personalized diagnostic profiles and facilitates more accurate judgment regarding the necessity and urgency of further investigations, such as cardiac MRI, CT, or genetic testing.

In the future, we aim to expand our research to include more sports and to prospectively validate the eligibility of Semmelweis Athlete’s Heart Calculator in a large number of healthy athletes and individuals with specific pathologies.

## 6. Patents

**Limitations:** Our study was based on the measurements of elite water polo athletes between the ages of 14 and 37. The obtained results cannot be applied directly to elite athletes from other sports or to the entire population. However, our results may be useful when screening members of similar mixed sports, as, until now, no sport- and age-specific recommendations have been made available in the literature.

Body composition was assessed using a bioimpedance-based (BIA) method, which, while generally reliable, is less accurate compared to other techniques like dual-energy X-ray absorptiometry (DEXA). However, given that BIA provides reasonably accurate results, does not involve radiation exposure, and is easily available and reproducible, it was chosen as the preferred method for this study. In future research, we plan to conduct DEXA-based assessments as well, where we expect to obtain comparable results.

In a subset of cases, the recommended 24 h period without training prior to the bioimpedance-based measurements was not fulfilled; some athletes had participated in morning training sessions on the day preceding the examination. However, a minimum of 18 h had elapsed between the end of the last training session and the time of assessment in all cases.

## Figures and Tables

**Figure 1 sports-13-00180-f001:**
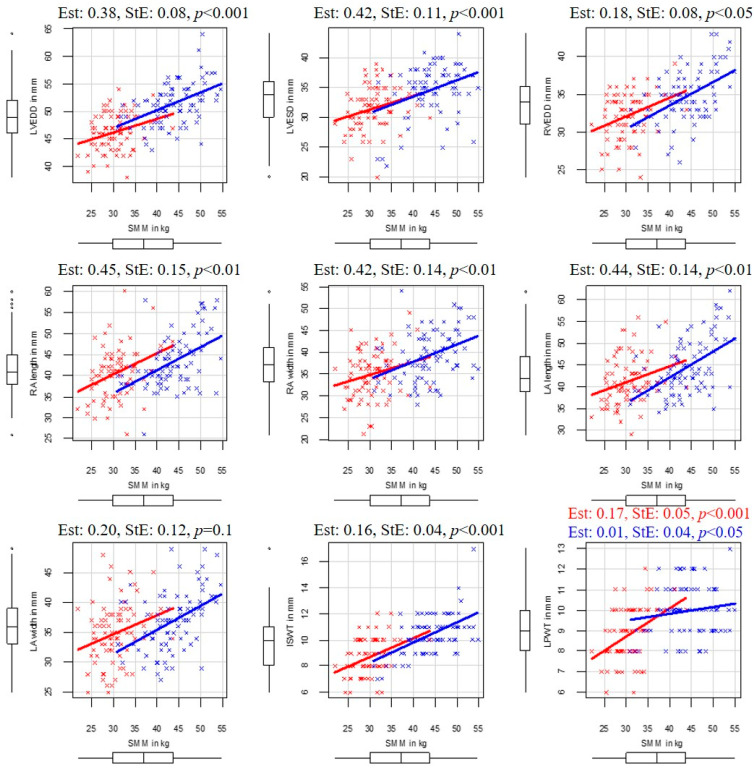
Correlation between SMM and the resting basic echocardiographic parameters of the examined elite water polo athletes. Abbreviations: ISWT—interventricular septal wall thickness; LA lengths—left atrium lengths; LA width—left atrium width; LPWT—left ventricular posterior wall thickness; LVEDD—left ventricular end diastole ic diameter; LVESD—left ventricular end systolic diameter; RA lengths—right atrial length; RA width—right atrial width; RVEDD—right ventricular end diastolic diameter; Est—Estimate; StE—standard error; SMM—skeletal muscle mass. Men are marked in blue and women in red.

**Figure 2 sports-13-00180-f002:**
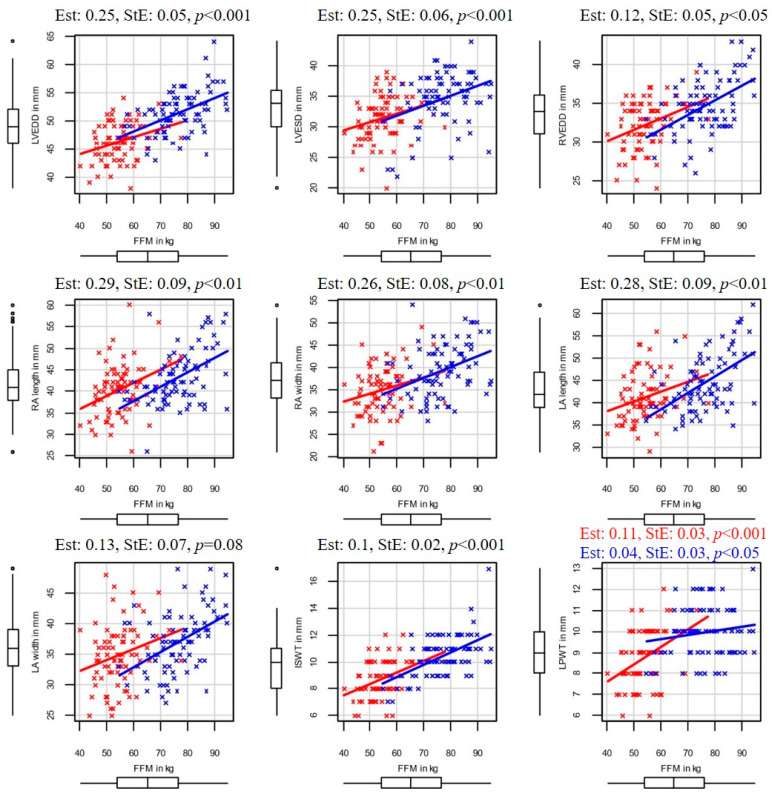
Correlation between FFM and the resting basic echocardiographic parameters of the examined elite water polo athletes. Abbreviations: ISWT—interventricular septal wall thickness; LA lengths—left atrium lengths; LA width—left atrium width; LVEDD—left ventricular end diastolic diameter; LVESD—left ventricular end systolic diameter; LPWT—left ventricular posterior wall thickness; RA lengths—right atrial length; RA width—right atrial width; RVEDD—right ventricular end diastolic diameter; FFM—skeletal muscle mass. Men are marked in blue and women in red.

**Table 1 sports-13-00180-t001:** Anthropometric parameters and body composition summary of the examined elite water polo athletes by sex and age.

Variable	Youth Male	Adult Male	Youth Female	Adult Female
Number (n)	54	28	55	24
Age (year)	**17.0 ± 1.2 **”**	**27.5 ± 4.8 ****	**15.7 ± 1.4 ##”**	**25.8 ± 3.8 ##**
Training (hours/week)	**16.1 ± 4.8 ****	**20.8 ± 5.1 **††**	**16.5 ± 5.6 ##**	**25.1 ± 5.3 ##††**
Weight (kg)	**82.0 ± 10.6 **”**	**98.3 ± 9.6 **††**	**66.8 ± 8.1 ##”**	**76.6 ± 10.2 ##††**
Height (cm)	**186.0 ± 6.3 **”**	**194.6 ± 4.9 **††**	**171.2 ± 5.7 ##”**	**178.8 ± 6.3 ##††**
SMM (kg)	**41.5 ± 4.7 **”**	**48.2 ± 3.6 **††**	**28.8 ± 2.7 ##”**	**34.1 ± 3.9 ##††**
FFM (kg)	**72.5 ± 7.8 **”**	**84.5 ± 6.2 **††**	**51.9 ± 4.6 ##”**	**60.6 ± 6.9 ##††**
BFM (kg)	**9.5 ± 4.2 **”**	**13.8 ± 5.6 ****	**15.0 ± 4.5 “**	15.9 ± 5.4
PBF (%)	**11.3 ± 3.6 **”**	**13.8 ± 4.2 **††**	**22.0 ± 4.6 “**	**20.5 ± 5.1 ††**

Abbreviations: BFM—body fat mass; FFM—fat-free mass; PBF—percent body fat; SMM—skeletal muscle mass. Significances: ††—adult male vs. adult female < 0.01; **—youth male vs. adult male < 0.01; ##—youth female vs. adult female < 0.01; “—youth male vs. youth female < 0.01.

**Table 2 sports-13-00180-t002:** Resting basic echocardiographic parameters of the examined elite water polo athletes by sex and age.

Variable	Youth Male	Adult Male	Youth Female	Adult Female
LVEDD (mm)	**50.4 ± 3.4 **”**	**53.1 ± 4.0 **††**	**46.3 ± 3.1 “**	**46.1 ± 3.9 ††**
LVESD (mm)	**34.3 ± 5.2 “**	**35.0 ± 4.3 ††**	**31.2 ± 3.8 “**	**32.0 ± 3.1 ††**
RVEDD (mm)	**33.7 ± 3.2 **”**	**36.7 ± 3.2 **††**	**31.8 ± 2.8 “**	**33.3 ± 3.5 ††**
RA lengths (mm)	**37.6 ± 5.4 **”**	**42.6 ± 5.5 **††**	**34.3 ± 4.8 “**	**36.3 ± 6.3 ††**
RA width (mm)	**41.4 ± 5.5 ****	**47.2 ± 6.3 **††**	39.7 ± 5.7	**41.9 ± 6.1 ††**
LA lengths (mm)	**34.9 ± 4.1 ****	**40.9 ± 4.3 **†**	**33.7 ± 4.5 ##**	**37.2 ± 5.3 ##†**
LA width (mm)	**42.6 ± 5.2 ****	**48.6 ±7.1 **††**	41.5 ± 5.1	**41.5 ± 6.1 ††**
ISWT (mm)	**9.9 ± 1.8 **”**	**11.0 ± 1.7 **††**	**8.5 ± 1.4 ##”**	**9.4 ± 1.3 ##††**
LPWT (mm)	**10.3 ± 1.1 **”**	**9.4 ± 1.2 ****	**8.6 ± 1.2 ##”**	**9.3 ± 1.4 ##**

Abbreviations: ISWT—interventricular septal wall thickness; LA lengths—left atrium lengths; LA width—left atrium width; LVEDD—left ventricular end diastolic diameter; LVESD—left ventricular end systolic diameter; LPWT—left ventricular posterior wall thickness; RA lengths—right atrial length; RA width—right atrial width; RVEDD—right ventricular end diastolic diameter. Significances: †—adult male vs. adult female < 0.05; ††—adult male vs. adult female < 0.01; **—youth male vs. adult male < 0.01; ##—youth female vs. adult female < 0.01; “—youth male vs. youth female < 0.01.

## Data Availability

The data that support the findings of this study are available from the corresponding author, M.Z., upon reasonable request. The data are not publicly available due to privacy and ethical restrictions.

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
