# Peer review of "Investigation of Body Composition and Cardiac Sports Adaptation in Elite Water Polo Players"

_sports, 2025, doi:10.3390/sports13060180_

Round 1
Reviewer 1 Report
Comments and Suggestions for Authors
Dear Authors,
The work submitted to me for evaluation is very valuable. The conducted study has an implication dimension, which is crucial in the diagnostics of the tool. The work requires extensions, corrections and supplements in individual sections:
material and methods:
in this section, the gradual course of the research experiment should be taken into account (taking into account the number, time ranges, and detailed studies)
- ask to highlight the criteria for inclusion and exclusion from the study (also placing them on the above graph)
- in this section, please also describe what method was used to select the research group
- please indicate how many people were rejected from the study, or were the only criteria pathological results of the parameters tested?
Results:
- in tables, please highlight statistically significant results
- it is tempting to present the results of the described correlations in graphical form - charts
- please add a section in the paper with limitations and describe in it the ones that occurred during the implementation of this research
- please add a link to a website with a free calculator in section 3.10 in the paper, which will facilitate quick access to this tool
best regards
Reviewer 2 Report
Comments and Suggestions for Authors
General comments
Could the relationship between body composition and cardiac dimensions be explained by player’s genetic and/or training/fitness level? This issue could be central for better interpreting results and highlight interesting and useful practical applications. Therefore, this issue should be treated in Discussion and/or Conclusion as limitation and/or further research purposes.
Specific comments
Abstract
Considering the part dedicated to Results, main finding should be reported. Practically, sex and age differences should be mainly reported.
Introduction
In the last part of this section, the lack of literature and the consequent reporting of experimental aims should be better defined and recognizable (“However, literature lacks of….. Therefore, the aims of this study were:…..”).
Materials and Method, and Results
These sections have been well structured and written
Discussion
The beginning of this section could be reduced for remarking introduction of the research theme and aims, providing immediate awareness about the main findings (age and sex differences).
Some notes about and speculated relationships between the findings of the study and requests of water polo performance could make this article as more readable and pertinent to the scopes of Sports journal. Although very few articles reported HR responses during water polo performance (Hollander et al., 1994; Lupo et al., 2015), these results could be useful to substantially make associations with the cardiac findings of the current study. Hollander, et al. (1994). Physiological strain during competitive water polo games and training. Medicine and Sport Science, 39, 178-178. Lupo, C., et al. (2015). Tactical swimming activity and heart rate aspects of youth water polo game. The Journal of sports medicine and physical fitness, 56(9), 997-1006.
Reviewer 3 Report
Comments and Suggestions for Authors
Abstract
Please indicate body composition values of the participants.
Introduction
The introduction is well structured and provides an adequate state of the art and points out the scientific niche.
Methodology
Please provide a section on the study design and indicate which experimental protocol was followed.
At what time did the assessments take place?
Please indicate the number of participants, age range, level of training of the participants. Please indicate what were the inclusion and exclusion criteria for the study.
In the body composition analysis, were the participants instructed not to exercise during the previous 24 h? During the bioimpedance measurement it is recommended that participants urinate at least half an hour before, was this done?
Was the diet of the participants controlled during the week of measurements?
Discussion
Please add a paragraph on practical applications for trainers and physicians.
Add limitations of the study, such as the use of bioimpedance instead of DXA.
Round 2
Reviewer 3 Report
Comments and Suggestions for Authors
The authors responded appropriately to my comments.